# In-air fast response and high speed jumping and rolling of a light-driven hydrogel actuator

Mingtong Li[1,2,3], Xin Wang[1,3], Bin Dong [1]✉ & Metin Sitti [2]✉

Stimuli-responsive hydrogel actuators have promising applications in various fields. However, the typical hydrogel actuation relies on the swelling and de-swelling process caused by osmotic-pressure changes, which is slow and normally requires the presence of water environment. Herein, we report a light-powered in-air hydrogel actuator with remarkable performances, including ultrafast motion speed (up to 1.6 m/s), rapid response (as fast as 800 ms) and high jumping height (~15 cm). The hydrogel is operated based on a fundamentally different mechanism that harnesses the synergetic interactions between the binary constituent parts, i.e. the elasticity of the poly(sodium acrylate) hydrogel, and the bubble caused by the photothermal effect of the embedded magnetic iron oxide nanoparticles. The current hydrogel actuator exhibits controlled motion velocity and direction, making it promising for a wide range of mobile robotics, soft robotics, sensors, controlled drug delivery and other miniature device applications.

[1] Institute of Functional Nano & Soft Materials (FUNSOM), Jiangsu Key Laboratory for Carbon-Based Functional Materials & Devices, Soochow University, 215123 Suzhou, Jiangsu, P. R. China. [2] Physical Intelligence Department, Max Planck Institute for Intelligent Systems, 70569 Stuttgart, Germany. [3] These authors contributed equally: Mingtong Li, Xin Wang. ✉email: bdong@suda.edu.cn; sitti@is.mpg.de

Soft actuators, which can convert external environmental stimuli into mechanical energy[1], have allured increasing interest recently due to their promising applications in sensors[2], robots[3–6], artificial muscles[7], medicine[8,9], etc[10–13]. Recently, aiming at achieving robust actuation with adaptable manipulation in complicated environments, several kinds of soft actuators have been designed, which include liquid crystal elastomer actuators[14–16], composite actuators[3,4,17,18], hydrogel actuators[19–22], etc[11,23]. Among others, inspired by the motion of plants whose winding and curving are driven by the absorption or dehydration of water in the cells and tissues, hydrogel actuator has attracted more and more attention, which is capable of undergoing a deformation as large as 10 times of its own volume by changing the amount of water in its network[24–26]. This unique property, together with their wetness and biocompatibility, makes the hydrogel actuators particularly useful in applications ranging from smart chemical valves[10,27], tissue scaffold engineering[28–30] to chemical reaction switches[31], and drug delivery vehicles[32,33].

In recent years, various types of hydrogel actuators have been developed which are able to walk[34,35], crawl[36,37], roll[38,39], and grasp[40] in respond to external stimuli, which include chemicals[41], temperature[42–44], electricity[45,46], humidity[47–49], and light[50–53]. The actuation of a hydrogel actuator relies on the swelling and de-swelling process caused by the osmotic-pressure changes when exposed to these stimuli, which is slow and requires the presence of water environment[54]. Due to the slow diffusion of water in and out of the hydrogel networks, even a slight volume shift (about 10%) of the hydrogel actuator takes several minutes or even hours, which hinders its practical application scope[36]. It still remains a challenge to achieve both high-speed, fast response, and in-air motion for the hydrogel actuators.

In this paper, we propose a high performance in-air hydrogel actuator, which is composed of a binary iron oxide nanoparticle (IONP, i.e. $Fe_3O_4NP$) and poly(sodium acrylate) (PAANa) hydrogel composite. Interestingly, it exhibits ultrafast motion, which can reach 1.6 m/s speed and a superior jumping height of 15 cm with fast response time of only 800 ms. The extraordinary performance of the current hydrogel actuator is realized by harnessing the synergetic interactions between the elasticity of the hydrogel and the bubble caused by the photothermal effect of the embedded IONPs. More interestingly, both jumping and rolling behaviors exhibit controllable moving velocities and directions, which make the hydrogel actuator reported in the current study attractive for various practical applications. As a proof-of-the-concept, we have demonstrated its potential use in the fields of sound-recording and cargo delivery robotics.

## Results

### Fabrication and characterization of the hydrogel actuator

The hydrogel actuator has a core shell structure, i.e., 60 μm-thick polybutadiene rubber (PBR) as the shell (whose function is to prevent the water loss from the hydrogel actuator to the surrounding environment) and the 1 mm-radius hydrogel core consisting of two components, i.e., IONPs and PAANa hydrogel (Fig. 1a–d). The hydrogel sphere is obtained based on the suspension polymerization method, i.e., free-radical polymerizing the aqueous droplet containing monomer, initiator, crosslinker, crosslinking accelerator and IONPs that suspends in liquid paraffin. After polymerization, washing, and drying, we have characterized the hydrogel sphere without the PBR surface coating. As can be seen from the scanning electron microscopy (SEM) image shown in Fig. 1e, the hydrogel has a rough surface with a surface root mean square (RMS) roughness of ~4.9 μm as characterized by the atomic force microscopy (AFM) measurement (Supplementary Fig. 1). The corresponding energy-dispersive X-ray (EDX)

analysis indicates that carbon (Supplementary Fig. 2) and iron (Fig. 1f) elements uniformly distribute throughout the whole structure, confirming the elemental composition. In addition, as shown in the Fourier transform infrared spectroscopy (FTIR) spectrum of the PAANa hydrogel and IONPs (Fig. 1g pink curve), the characteristic peaks of $Fe_3O_4$ (522 cm$^{-1}$ and 1550 cm$^{-1}$ attributed to the Fe–O bonds stretching vibrations and the bending vibrations of N–H bonds originating from the oleamine surface ligand (Fig. 1g yellow curve)) and PAANa (1410 cm$^{-1}$ assigned to COO$^-$ symmetric stretching (Fig. 1g blue curve)) are all visible, thus further indicating the presence of the PAANa and IONP components in the hydrogel structure[55,56]. Furthermore, since the IONPs are positively charged (measured to be 18 mV) due to the oleamine surface ligands and PAANa hydrogels are negatively charged (measured to be −24 mV), there may be electrostatic interaction between these two constituent parts (as illustrated in Fig. 1c). Moreover, the resulting IONPs and PAANa composite hydrogel displays a broad absorption band peaked at 850 nm (as shown in the UV-Vis-NIR spectrum indicated in Supplementary Fig. 3), which originates from the absorption of IONPs. This indicates that IONPs may be capable of absorbing the light energy (preferably the NIR light) and converting it into the thermal energy, which is in consistent with literature results[57]. As a consequence, the hydrogel actuator may be responsive to NIR due to the photothermal effect of the IONPs and the local changes in temperature near the irradiated area.

### Actuation of the hydrogel actuator

The current hydrogel actuator could move in-air upon the light actuation. There are basically two different motion behaviors, i.e., jumping and rolling, which depend on the irradiation position on the surface of the hydrogel actuator. Among others, when irradiating the bottom part, the hydrogel actuator exhibits the jumping motion, as shown in Fig. 1h and Supplementary Video 1. The jumping features fast response (800 ms) with ultrafast take-off speed of around 1.6 m/s under 2.34 W irradiation obtained through the jumping trajectory analysis shown in Fig. 1i–j. We have compared the response and motion speed with those of other hydrogel actuators. As summarized in Supplementary Table 1, the current hydrogel possesses a much higher motion velocity than the hydrogel actuators reported to date with a faster response time. The fast response, together with ultrafast take-off process renders the current hydrogel actuator superior jumping performance, i.e., it can achieve a jumping height as high as 15 cm (Supplementary Fig. 4a), which is by far the highest value for hydrogel actuator-based jumper[17,42,49]. On the other hand, when irradiating the side of the hydrogel actuator, it exhibits the rolling motion, as shown in Fig. 1k and Supplementary Video 2, with response time of 1.3 s. As can be seen from Fig. 1l–m and Supplementary Fig. 4b, the rolling velocity is estimated to be around ~10 cm/s under 0.67 W laser through the trajectory analysis and the actuator is able to reach to a distance of ~20 cm because of the little resistance of the sphere-like structure on the smooth glass substrate during the rolling process. In addition, we have analyzed the velocity changes after the light-actuated jumping and rolling. As shown in Supplementary Fig. 5, the speed of the hydrogel after actuation gradually decreases.

### Actuation mechanism of the hydrogel actuator

The light-driven jumping and rolling behavior of the hydrogel actuator is likely because of the shape deformation caused by the bubble formation due to the photothermal effect of IONPs, as illustrated in Fig. 2a. Among others, the mechanism of the jumping behavior is studied by recording the take-off process using the high-speed camera, the thermal imaging and the computer simulation. As

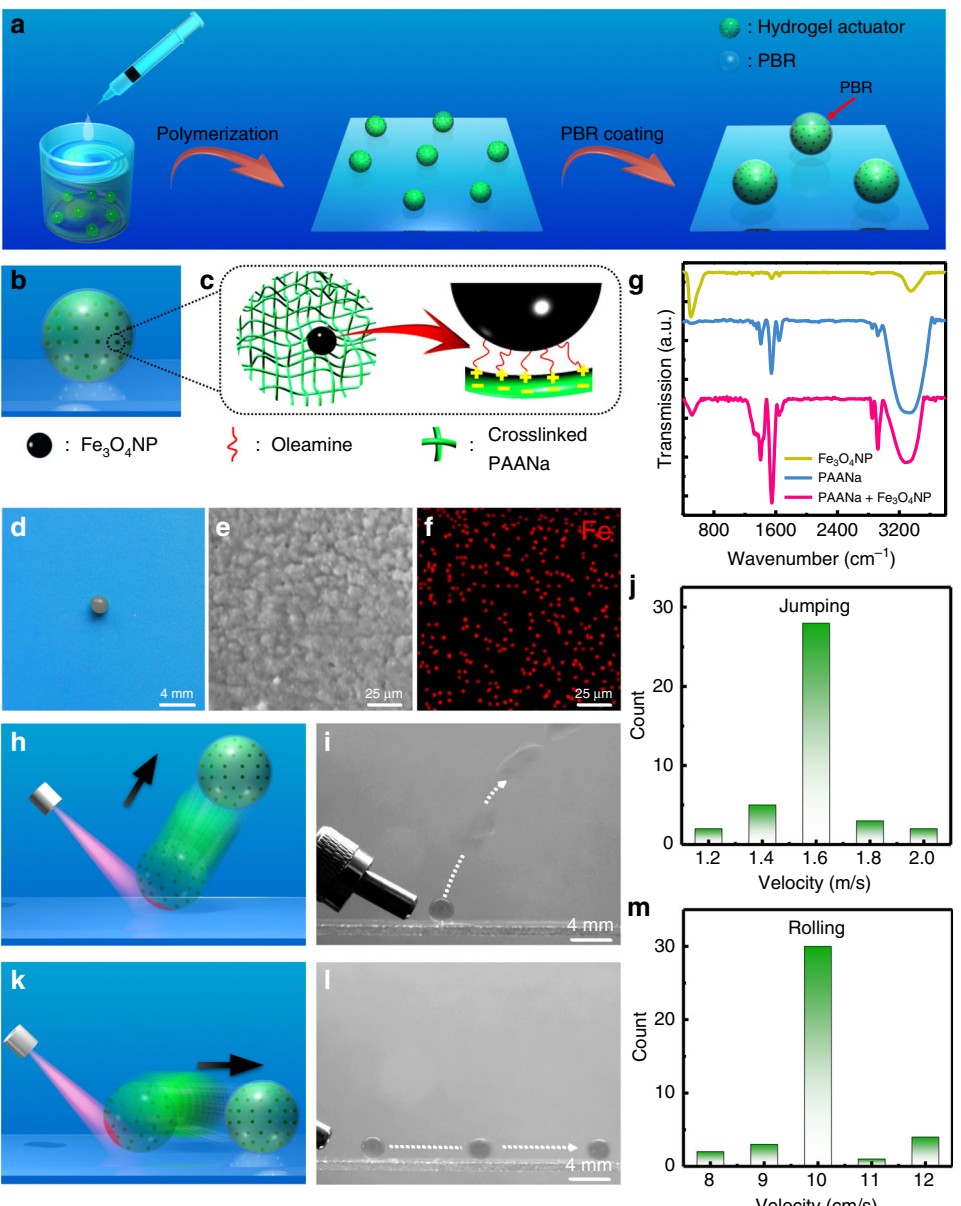

**Fig. 1 The hydrogel actuator and its light-induced jumping and rolling motion in-air. a** The fabrication process of the hydrogel actuator. **b** Schematic illustrating the hydrogel actuator. **c** Schematic showing the internal structure of the hydrogel actuator. **d** The typical CCD image of the resulting hydrogel actuator. **e** SEM image indicating the surface morphology of the hydrogel actuator, and **f** the corresponding EDX analysis for the iron element. **g** The FTIR spectra of the PAANa hydrogel (blue curve), IONPs (yellow curve) and the hydrogel actuator consisting of both PAANa hydrogel and IONPs (pink curve), respectively. The schematics, overlaid CCD images and the take-off velocity showing **h–j** jumping (under 2.34 W irradiation) and **k–m** rolling (under 0.67 W irradiation) behavior of the hydrogel actuator. (i) and (l) are obtained from Supplementary Video 1 and 2, respectively. For jumping and rolling, light is applied at the bottom part or the side of the hydrogel actuator (the red area in (**h**) and (**k**)), respectively.

shown in Fig. 2b–e and Supplementary Video 3, there is a rapid expansion of the illuminated part in the shape of a bubble within 3.2 ms (from Fig. 2b–d) before the hydrogel actuator is airborne, indicating it is this fast deformation that accounts for the following motion. We have recorded the temperature change during the motion process. As can be seen from Fig. 2f–h, the temperature of the hydrogel actuator raises from 22 °C before light irradiation to 126 °C (a temperature beyond the boiling point of water (i.e., 100 °C)) under light irradiation. The rapid temperature increase originates from the photothermal effect of IONPs and causes the gasification of water and bubble formation. The generation of the gas bubbles is further demonstrated by the diving and surfacing experiment shown in Supplementary

Fig. 6 and Video 4. The hydrogel actuator which has a density of ~1.9 g/cm³ dives to the bottom of the N-methylpyrrolidone solvent (1.4 g/cm³) in the absence of light and surfaces to the top under the light irradiation, indicating the density becomes lower than 1.4 g/cm³ which is likely caused by the formation of the gas bubbles during the illumination process. In addition, the shape deformation and recovery process could also be directly observed when fixing the hydrogel actuator. As shown in Fig. 2i–l and Supplementary Video 5, the local volume of the hydrogel actuator swells at the irradiated spot due to the bubble formation and restores to its original state after the temperature decreases within 8 s, which correlates well with the temperature change (Fig. 2h) during the shape changing process. Furthermore, as shown in

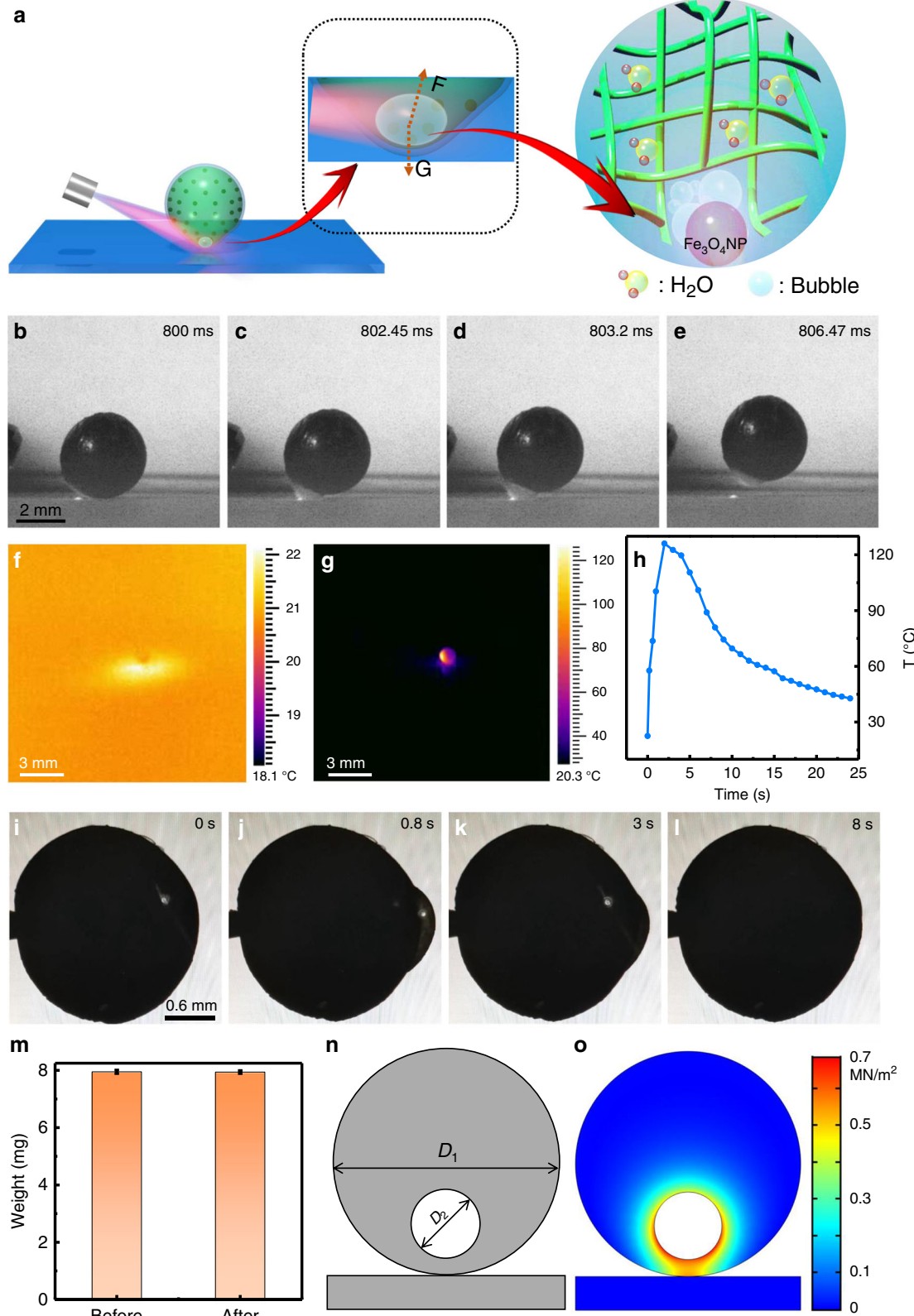

**Fig. 2 Motion mechanism of the hydrogel actuator. a** Schematic illustrates the local shape change of the hydrogel actuator caused by the photothermal effect of IONP and the thereafter water vaporization under light irradiation. **b–e** A series of high-speed camera images obtained from Supplementary Video 3 showing the take-off process of the hydrogel actuator under light irradiation (2.34 W) at the bottom part of the hydrogel actuator. Infrared thermal images of the hydrogel actuator **f** before and **g** after light irradiation. **h** The temperature change during the jumping process. **i–l** A series of CCD images indicating the shape deformation and recovery at the irradiated area of the hydrogel actuator fixed at a needle end. **m** The weight of the hydrogel actuator before and after jumping. **n–o** The force analysis of the jumping behavior of the hydrogel actuator by using the numerical simulations. Error bars denote the standard deviation.

Fig. 2m and Supplementary Fig. 7, since there are almost no changes of either the weight or surface morphology, it indicates that the water vapor is not released from the hydrogel actuator to the surrounding environment due to the presence of the PBR surface coating. Given the fact that there is still deformation in the case of the uncoated hydrogel/IONP actuator (without PBR outer layer) under light irradiation (Supplementary Fig. 8 and high-speed camera movie (Video 6)) with much faster recovery time (within 3 ms), this shape recovery of the PBR coated actuator may be attributed to the resorption of water by the hydrogel and the good elasticity of the hydrogel and PBR. Based on the above observation, we propose the following tentative mechanism: Under light irradiation, the temperature of the hydrogel actuator rises rapidly due to the photothermal effect of IONPs, leading to the vaporization of water at the irradiated area inside the hydrogel, rapidly forming gas bubble. The fast expanding local protrusion hits the substrate and the counter-interaction provides the driving force for the jumping behavior.

We have analyzed the propulsion force of the hydrogel actuator in the vertical jumping process based on the Newtonian equation as follows:

$$F_{jumping} = mv_j/t_j + G \tag{1}$$

where $F_{jumping}$ is the driving force for jumping; $G$ is the gravitational force, which is 78 μN for the current hydrogel actuator; $v_j$ and $t_j$ are the jumping velocity and deformation time of the hydrogel actuator, respectively; $m$ is the mass of the hydrogel actuator. The propulsion force of the hydrogel actuator is calculated based on equation (1) to be ~4.05 mN, which is higher than the gravitational force of the actuator (i.e., 78 μN). In addition, we have developed a model consisting of a circle with a diameter of $D_1 = 2$ mm and a bubble cavity with a diameter of $D_2 = 0.6$ mm according to the experiment results (Fig. 2n). The driving force in the vertical jumping case ($F_{jumping}$) is related to the contact pressure ($\sigma$) between the actuator and the substrate according to the following equation:

$$F_{jumping} = \sigma S \tag{2}$$

where $S$ is the contact area between the hydrogel actuator and the substrate; $\sigma$ can be obtained through the numerical simulation (Fig. 2o and Supplementary Fig. 9 IV black curve), which is related to the pressure difference ($\Delta P$) inside ($P$) and outside ($P_0$) the bubble, i.e., $\Delta P = P - P_0$, where $P_0$ is the atmospheric pressure and $P$ is the pressure in the bubble cavity produced by the water vapor inside the hydrogel actuator. $P$ can be expressed by the following equation[58]:

$$P = P_c \times \exp(T_c/T \times (a_1\tau + a_2\tau^{1.5} + a_3\tau^3 + a_4\tau^{3.5} + a_5\tau^4 + a_6\tau^{7.5})) \tag{3}$$

where $P_c = 22.064$ MPa and $T_c = 647.096$ K are the critical pressure and temperature, respectively; $T$ is the actual temperature of the hydrogel actuator; $\tau = 1 - T/T_c$; $e$ and $a_{1-6}$ are the constants. As a result, the driving force for the take-off process of the hydrogel actuator could then be estimated based on Eqs. (2) and (3) to be ~11.34 mN, which is on the same order of magnitude as the force calculated based on Eq. (1) (4.05 mN), indicating that the local shape deformation is likely to be the driving force for jumping.

On the other hand, the rolling process also follows the similar mechanism as can be seen from Supplementary Fig. 10 and Video 7 where the bubble formation process is clearly visible except that the deformation and resulting driving force occurs at a different location. The propulsion force for the rolling motion is calculated to be around 0.1 mN according to the following equation: $F_{rolling} = mv_r/t_r$, where $v_r$ and $t_r$ are the rolling velocity

and deformation time of the hydrogel actuator, respectively. According to the numerical simulations (Fig. 2o), the maximum strain of the hydrogel actuator during bubble formation is estimated to be around 20%. This strain is lower than the fracture strain of the hydrogel comprising the actuator (i.e., >42%, Supplementary Fig. 11). This, together with the negligible changes in the surface morphology of the hydrogel comprising the actuator before and after NIR actuation (Supplementary Fig. 12) and negligible water loss after the motion process (Fig. 2m), indicates the hydrogel does not fracture during the bubble formation process, which also accounts for the reusability of the hydrogel actuator (more than 100 times repetitive actuation, Supplementary Fig. 13).

**Motion control of the hydrogel actuator**. Moreover, the moving (jumping and rolling) velocities of the hydrogel actuator are highly dependent on the content of $H_2O$, the content of IONP and the size of the hydrogel actuator. Among others, as shown in Supplementary Fig. 14a, d, the hydrogel actuator with less $H_2O$ content could only generate small bubbles due to the limited amount of water vapor produced under irradiation, thus exhibiting reduced moving velocity. However, if the $H_2O$ content is too high, the excess amount of water also lowers the heating efficiency caused by the photothermal effect of IONP, leading to less vapor formation, smaller bubbles, and slower motion. A medium $H_2O$ content of 60 wt% facilitates the best motion performance of the hydrogel actuator. In addition, as shown in Supplementary Fig. 14b, e, the amount of IONPs must be high enough to generate sufficient photothermal effect to propel the motion of the hydrogel actuator. However, when the IONP content is too high, the Young's modulus of the resulting hydrogel increases dramatically (as shown in Supplementary Fig. 15), reducing its flexibility and causing smaller bubble-induced deformation of the hydrogel actuator. The optimal IONP content is evaluated to be around 2 wt%. Besides, we have studied the motion velocity of different sized hydrogel actuator. As can be seen from Supplementary Fig. 14c, f, the optimal radius is 1 mm. For smaller actuator, the deformation at the irradiation area is also small due to the lower water content at the irradiation area, which cannot propel the hydrogel actuator at a high speed; while for larger actuator, the increase in the weight of the hydrogel actuator will suppress the increase of the propulsion force, leading to the decrease in the movement speed as well. The optimum water, IONP content and radius of actuator are thus 60 wt%, 2 wt %, and 1 mm, respectively, in order to achieve the best movement performance.

Different motion behaviors (jumping or rolling) of the hydrogel actuator could be realized by adjusting the laser power and irradiation position, i.e., rolling only between 0.04 and 0.39 W, jumping or rolling between 0.39 and 0.67 W depending on the irradiation position and jumping only between 0.67 and 2.34 W. Note that the bubble-induced protrusion needs to have direct contact with the substrate in order to realize any motion in the case of both jumping and rolling (Supplementary Fig. 16). Among others, at the medium laser power (0.39–0.67 W), the motion behavior (rolling or jumping) is highly dependent on the irradiation position, as shown in the schematics and the force analysis in Supplementary Fig. 16. In the case of 0.67 W irradiation, the component force ($F_{dr}^{\perp}$) acting on the hydrogel actuator along the vertical direction can be expressed by the following equation: $F_{dr}^{\perp} = F_{dr} \times \cos^2(180° - 2\alpha)$. $F_{dr}$ could also be estimated based on $F_{jumping}$ under similar irradiation condition from Eq. (1), which is calculated to be around 0.24 mN; $\alpha$ is defined as the angle between $d$ (the virtual line between the irradiation position and the bottom of the actuator) and $Z$-axis as

illustrated in Supplementary Fig. 16. In the case that $F_{dr}^{\perp}$ equals to the gravitational force of the hydrogel actuator (i.e., 78 μN), $\alpha$ could be calculated to be ~62.4°. Since $F_{dr}^{\perp}$ has to be smaller than the gravitational force for the rolling process, $\alpha$ should be smaller than 62.4°. However, since the generated bubbles normally has a diameter smaller than 0.8 mm, $\alpha$ has to be larger than 58° (calculation shown in Supplementary Fig. 16) so that the bubble-induced protrusion could contact the substrate in order to provide the driving force. Therefore, the hydrogel actuator rolls when $58° \leq \alpha \leq 62.4°$ and jumps when $62.4° < \alpha \leq 90°$. The experiment results show that the actuator rolls when $58° \leq \alpha \leq 63°$ and jumps when $63° < \alpha \leq 90°$ under 0.67 W irradiation, which correlates well with the calculation shown above, indicating the good controllability. In addition, as shown in Supplementary Fig. 17, the critical value of $\alpha$ between rolling and jumping is highly dependent on the light irradiation with decreasing $\alpha$ at higher laser powers. On the other hand, the rolling only motion is observed at low laser power (0.04–0.39 W), which is because the slow bubble formation cannot provide sufficient driving force for jumping. Also, at the high laser power regime (0.67–2.34 W), the component driving force along the vertical direction suppresses the gravitational force so that the actuator always jumps. Note that $\alpha$ in these two cases is limited to $58° \leq \alpha \leq 90°$ in order to realize the movement of the hydrogel actuator. In addition, the energy and power densities of jumping and rolling motion are estimated under different circumstances, as detailed in the 'methods' section.

Furthermore, as can be seen from Fig. 3 and Supplementary Video 8–11, the direction of the jumping and rolling behaviors can also be manipulated by adjusting the irradiation position (the red area in Fig. 3c, f, j, m inset). Figure 3g, n indicates the calibration curve of the moving speed of the hydrogel actuator versus the laser power. It can be seen that there is a nearly linear relationship between the laser power and the moving velocity of the hydrogel actuator, indicating the well-controlled maneuverability of the current hydrogel actuator. The moving speed increases monotonically from around 0.02 m/s (jumping) and 0.44 cm/s (rolling) to 1.6 m/s (jumping) and 10 cm/s (rolling) by increasing the laser power from 0.39 to 2.34 W (jumping) and from 0.04 to 0.67 W (rolling), respectively, which are mainly due to the faster bubble formation caused by the higher heating effect under stronger light irradiation (Supplementary Fig. 18). Note the hydrogel actuator burns when the light illumination is too strong, i.e., higher than 2.34 W. Moreover, since the jumping height and the rolling distance are all closely related to the motion velocity of the hydrogel actuator, the laser power could also be utilized to adjust the jumping height and rolling distance in a quantitative manner, as illustrated in Supplementary Fig. 19.

Furthermore, the actuator's trajectories and destinations can be controlled (Supplementary Fig. 20) for both jumping and rolling behaviors when laser power and $\alpha$ are fixed. For example, the hydrogel actuator exhibits controllable jumping (Supplementary Fig. 21 and Video 12) and rolling (Supplementary Fig. 22 and Video 13) trajectory and destination under 0.39 W light irradiation/$\alpha = 70°$ and 0.14 W light irradiation/$\alpha = 60°$, respectively. Also, different trajectories and destination of the actuator could be realized by tuning the laser irradiation position or the laser power. For instance, the actuator shows controllable jumping trajectories (take-off angle and the jumping height (Supplementary Fig. 23 and Supplementary Fig. 19a)) and destinations and controllable rolling destination (Supplementary Fig. 24 and Supplementary Fig. 19b) by adjusting $\alpha$ and laser power, respectively. Based on this accurate control, the motion (jumping and rolling) destinations of the hydrogel actuator can be predicted. Therefore, by moving the laser source to the

destination of the actuator after the first-time actuation, the second-time actuation could be achieved. As indicated in Supplementary Fig. 25 captured from Supplementary Video 14, we first actuate the hydrogel actuator to make it roll from position A to the designated position B (under 0.14 W irradiation). Then the laser is moved to position B to achieve the continuous actuation (jumping followed by rolling from position B to position C, under 0.39 W irradiation).

**Potential applications of the hydrogel actuator.** The fast response light-driven hydrogel actuator with jumping and rolling capabilities exhibits excellent applicability under different circumstances. As shown in Fig. 4a, e–g and Supplementary Video 15, the hydrogel actuator can jump over a barrier, which is 2 cm high and the maximum barrier height the hydrogel actuator can jump across is ~14 cm, which is slightly lower than the highest jumping height of the hydrogel actuator (15 cm). In addition, the hydrogel actuator is capable of rolling over an incline under light actuation. As shown in Fig. 4b, h–j and Supplementary Video 16, the hydrogel actuator can rapidly roll over a tilted substrate with an incline angle of 6°. The maximum angle the current hydrogel actuator can roll over is ~15°. Furthermore, based on the good controllability of the light-driven rolling behavior, the hydrogel actuator can be continuously guided toward a narrow slit which is only 1.6 mm wider than the hydrogel actuator and go through it, as illustrated in Fig. 4c, k–m and the corresponding Supplementary Video 17. The light-assisted actuation is an untethered propulsion method suitable for actuating the hydrogel actuator in a small and enclosed space with precision, which cannot be achieved by manual operation. Interestingly, the hydrogel actuator is capable of exhibiting the complicated combined rolling and jumping behavior when moving in a complex environment, e.g., a pellucid Z-pipe. As shown in Fig. 4d, n–s and Supplementary Video 18, the hydrogel actuator first rolls to the corner of the Z-pipe. Surprisingly, it can then jump and adhere to the wall of the pipe repetitively and controllably before finally exiting the pipe, indicating its great versatility in motion behaviors and the ability to operate under different circumstances. Note that the adhesion between the actuator and pipe wall is likely due to the synergistic interactions between the stickiness of the PBR and the impact force caused by the high-speed jumping[59].

In recent years, small-scale mobile robotics (with sizes ranging from several hundreds of micrometers to several millimeters) have undergone tremendous progresses and hold great promises in the field of targeted drug delivery, minimally invasive diagnosis and treatment inside the body, biotechnology, manufacturing, and mobile sensor networks[60]. The small size, fast response, rolling, and jumping capability make the current actuator potentially attractive in different small-scale robotic applications. Particularly, as a proof-of-the-concept, we have explored its potential as a sound-recording robot (Supplementary Fig. 26a). To this end, we have designed a robot consisting of a mini detectaphone (containing a battery and operated by a smart phone) assembled with four wheels composed of the hydrogel actuators (Supplementary Fig. 26b), which has a dimension of ~$8 \times 4 \times 2$ mm and a total weight of about 10 g. Based on the robust, fast and accurate moving performance of the hydrogel actuator, the robot which is placed far away from the sound source (i.e., a speaker) can be operated, approaching the speaker and wirelessly recording the sound, as can be seen from the different relative positions shown in Supplementary Fig. 26c–e and the corresponding sound amplitude increase shown in Supplementary Fig. 26g–i. After monitoring, the robot can be driven away from the speaker under illumination (Supplementary

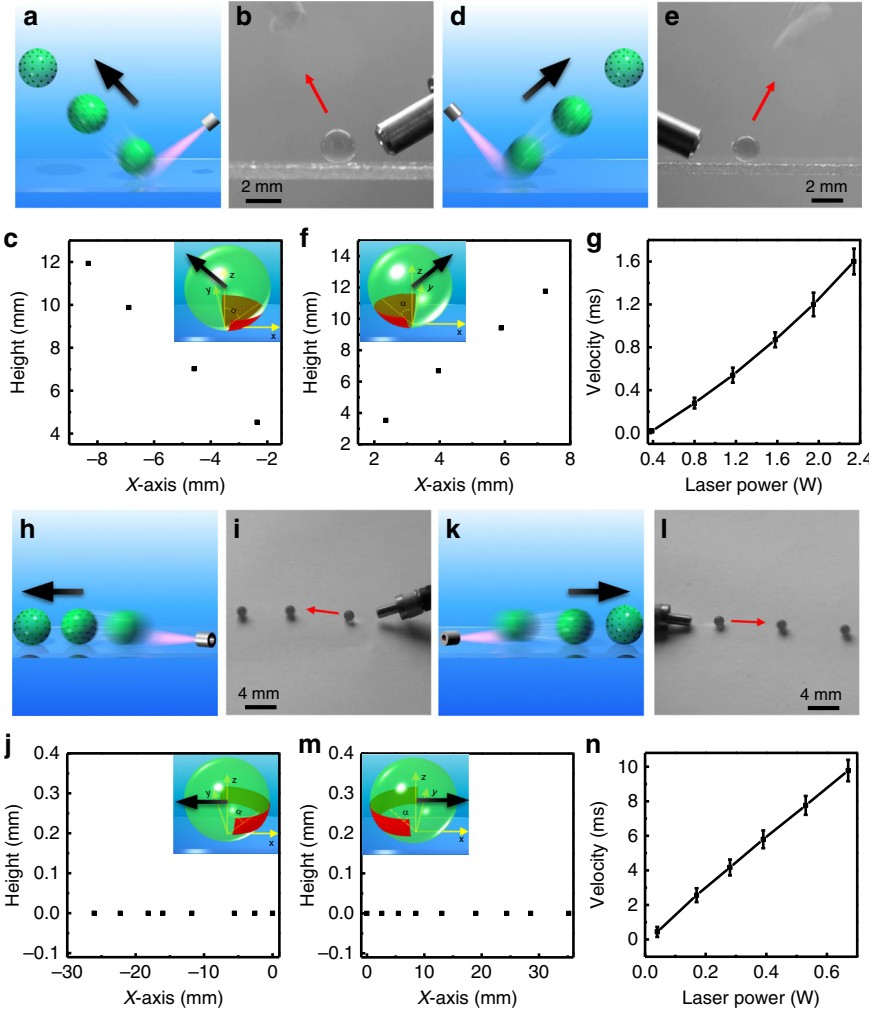

**Fig. 3 The motion control of the light-driven in-air hydrogel actuator.** Schematics, overlaid CCD camera images and moving trajectories indicating the motion of the hydrogel actuator, i.e., jumping to **a–c** left and **d–f** right under 2.34 W irradiation, rolling to **h–j** left and **k–m** right under 0.67 W irradiation, which are controlled by light irradiation position. Inset: Schematics illustrate the irradiated position (red area) in case of different motion behaviors. The laser power dependent jumping and rolling velocities are shown in **g** and **n**, respectively. The images in **b**, **e**, **i**, and **l** are captured from Supplementary Movies 8–11, respectively. Note that the left rolling or jumping distance is defined as negative. Error bars denote the standard deviation.

Fig. 26f), as evidenced by the reducing sound amplitude indicated in Supplementary Fig. 26j. The corresponding movie indicating the whole process is shown in Supplementary Video 19. This also paves the way toward directly mounting an NIR laser fiber (even a laser source) on the actuator so that the laser could move with the hydrogel actuator together, which may lead to the continuous actuation of the hydrogel actuator. Furthermore, the current hydrogel actuator is even capable of realizing controlled cargo transportation and delivery. As can be seen from Supplementary Fig. 27 and the corresponding movie shown in Supplementary Video 20, the actuator with partial surface attached with a model cargo consisting of paper due to the stickiness of PBR can still be driven inside a glass tunnel when irradiating the uncovered portion of the actuator surface, thus transporting the cargo together with it. Once the actuator arrives at the desired location, the model cargo could be controllably released by irradiating the position near the cargo covered actuator surface so that the light-induced protrusion decreases the contact area between the cargo and the actuator, leading to the cargo detaching from the actuator surface and thus cargo release (Supplementary Fig. 27).

Moreover, we have studied the actuation of the current hydrogel actuator from a longer operation distance and the actuation of bigger-sized ones, which may facilitate its further use

in practical applications. To this end, it is desirable to keep the irradiation intensity by narrowing the divergent NIR beam through the application of a collimator at the end of the NIR optical fiber (Supplementary Fig. 28). The NIR laser beam without a collimator is divergent so that the spot area on the hydrogel actuator is dependent on the distance from the laser to the object and the temperature of the hydrogel actuator is only higher than 100 °C when the distance between the NIR light and the actuator is less than 4 mm (with a spot area of approximately 0.28 mm² at 2 mm separation) as shown in Supplementary Fig. 29. Once the divergent NIR light is narrowed by the collimator, the local temperature can still exceed 100 °C even if the distance between the light source and the actuator is increased to 35 mm (Supplementary Fig. 30). As a result, the jumping and rolling behaviors of the hydrogel can be actuated by the NIR light (2.34 W) (Supplementary Fig. 31, Video 21 and 22) at longer operation distances (e.g., 35 mm), which is in contrast to <4 mm without using a collimator. It is anticipated that the actuation distance could be further increased when utilizing an NIR light source with higher power. In addition, the application of a collimator could help to generate bigger bubbles for bigger actuators. Since the driving force of the hydrogel actuator scales with the contact pressure (Eq. (2)), which depends on the size of

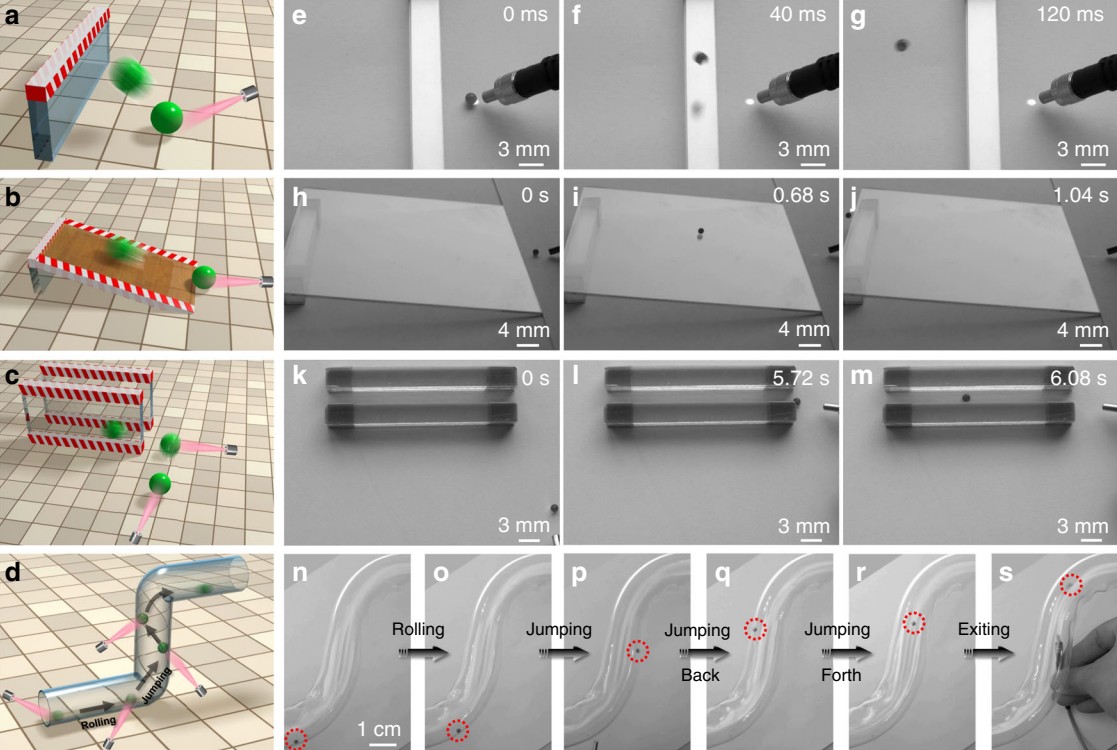

**Fig. 4 Schematic illustrates the applicability of the hydrogel actuator under different circumstances. a** jumping over a barrier, **b** rolling over an incline, **c** going across a slit, and **d** moving through a pellucid z-shaped pipe. The corresponding images of **a–d** are shown in **e–g**, **h–j**, **k–m**, and **n–s**, respectively. The images of **e–g**, **h–j**, **k–m**, and **n–s** are obtained from the Supplementary Movies 15–18, respectively.

the bubble cavity and the hydrogel actuator (Supplementary Fig. 9 and Supplementary Fig. 32), the larger bubble formation would provide a direct propulsion force for the jumping and rolling behaviors of large (e.g., 8 mm diameter) hydrogel actuators, as illustrated in Supplementary Fig. 33, Video 23 and 24.

## Discussions

A fast response (as fast as 800 ms) in-air jumping and rolling hydrogel actuator is driven by light with ultrafast speed (up to 1.6 m/s) and high jumping height (15 cm) by harnessing the synergetic interactions between the binary constituent parts, i.e. the elasticity of the hydrogel and the photothermal effect of the IONPs. The moving direction, velocity, height, and distance of the current hydrogel actuator can all be controlled by adjusting the irradiation position and laser power, which renders it versatile motion capabilities, including jumping across a barrier, rolling over an incline, going across a slit and jumping back and forth inside a pellucid Z-pipe or even functioning as a sound-recording or cargo delivery robot. The current hydrogel actuator with excellent durability steps away from the liquid environment generally required for operating a hydrogel actuator, which may not only stimulate new design principles for in-air hydrogel actuator but also have great potential in a wide range of robotics, biotechnology, and other miniature device applications. Specifically, the current actuator may be utilized for unclogging or repairing jammed or damaged pipes or tubes made of NIR-transparent materials, such as plastics and glasses, by releasing specific cargoes inside. Such industrial repair/maintenance application is possible for nuclear plants, space shuttles, airplanes, etc., where there is limited access to small pipes/tubes[61,62]. In addition, by utilizing biodegradable hydrogels, small-scale (e.g., 100 μm diameter, Supplementary Fig. 14f) hydrogel balls may achieve the active and precise delivery of drugs (due to its controlled motion behavior driven by external NIR light) for disease

or blood clot treatment after injection to the blood vessels close to skin (within ~10 mm depth) on legs, arms, or neck[63,64].

## Methods

**Materials.** Sodium acrylate, 30 nm IONP, N,N'-methylenebisacrylamide (MBA), potassium persulfate, N,N,N',N'-tetramethylethylenediamine (TEMED), PBR, sodium dodecyl sulphate (SDS), liquid paraffin, and chloroform were obtained from Sigma–Aldrich Company.

**Fabrication methods.** The hydrogel actuator was fabricated by the suspension polymerization method. In a typical experiment, 400 mg of sodium acrylate, 5 mg of SDS and 20 mg of MBA (the crosslinker) were first dissolved in 2 mL of deionized water. Fifty milligram of IONPs were then added into the solution and stirred for 5 min. Next, the above solution was purged with $N_2$ gas for 20 min. Five milligram of potassium persulfate and 20 μL of TEMED were added to the whole solution and stirred for 2 s. The resulting solution was the water phase, which was then dispersed in a liquid paraffin oil phase with 1:25 volume ratio by agitation, forming small droplets. Finally, after washing the precipitated hydrogel sphere by chloroform, the spherical hydrogel actuator was dried at 60 °C for 4 h and then coated with PBR by immersing it in the 25 wt% PBR's chloroform solution followed by drying. In the current study, unless otherwise mentioned, the water content, the IONP content and the radius of the hydrogel actuator were 60 wt%, 2 wt%, and 1 mm, respectively.

**Characterization.** The SEM measurement was performed on a Quanta 200 FEG (FEI Company) scanning electron microscope. The light source utilized in the current study was 808 nm NIR light from Hi-Tech Optoelectronics (LOS-BLD-0808-15W-C/P). Unless otherwise mentioned, the laser power used to actuate the jumping and rolling motion of the hydrogel actuator were 2.34 and 0.67 W, respectively. The AFM measurement was carried out on a Bruker Dimension scanning probe microscope at room temperature. A silicon tip was utilized to measure the surface morphology of the hydrogel actuator in-air. Young's modulus of the hydrogel comprising the actuator was obtained by utilizing a tensile tester (INSTRON-5942). SONY DSC-RX10III digital camera was used for capturing the CCD images. The high-speed videos were obtained by utilizing Phantom V310 camera. The moving velocity was obtained by analyzing the captured videos using the PhysVis software. The zeta potential of the IONPs and PAANa hydrogel was measured using Zetasizer Nano-ZS 90 (Malvern Instruments). FTIR was performed on the Bruker VERTEX 70 spectrometer. UV-Vis-NIR measurements were carried out on a PerkinElmer Lambda 750 spectrophotometer. The infrared thermal

images were obtained by using a FLIR-A300 camera (FLIR Systems Inc.). The numerical simulation was conducted based on the COMSOL Multiphysics software.

**Estimation of the actuator energy density and power density**. The light-actuated shape deformation caused by the bubble formation due to the photothermal effect of IONPs produces energy ($E_{produced}$) that generates jumping ($E_{jumping}$) or rolling ($E_{rolling}$) motion by overcoming the stiction force on the ground ($E_{stiction}$), the air drag ($E_{air\ drag}$), the internal energy dissipation ($E_{dissipation}$), and the friction between the actuator and the ground ($E_{friction}$)[49]. In case of jumping:

$$E_{produced} = E_{jumping} + E_{stiction} + E_{air\ drag} + E_{dissipation} \qquad (4)$$

In case of rolling:

$$E_{produced} = E_{rolling} + E_{stiction} + E_{air\ drag} + E_{friction} + E_{dissipation} \qquad (5)$$

The minimum produced energy could be estimated by $E_{jumping}$ (in case of jumping) or $E_{rolling}$ (in the case of rolling) because the produced energy ($E_{produced}$) is higher than that of jumping ($E_{jumping}$) or rolling ($E_{rolling}$). Among others, $E_{jumping}$ can be analyzed based on the Newtonian equation as:

$$E_{jumping} = 1/2 \times mv_j^2 \qquad (6)$$

where $m$ is the mass of the hydrogel actuator, which is 7.95 mg; $v_j$ is the take-off velocity. The energy density of the hydrogel actuator can thus be estimated by:

$$Energy\ density = E_{jumping}/m_{expanding} \qquad (7)$$

where $m_{expanding}$ is the mass of the actuation part of the actuator (~0.21 mg), which is calculated from the size of the local expanding part ($m_{expanding} = \rho \times V_{expanding}$, where $\rho$ is the density of the hydrogel actuator; $V_{expanding}$ is the volume of the local expanding part). Therefore,

$$Energy\ density = 1/2 \times mv_j^2/m_{expanding} \qquad (8)$$

where $v_j$ is dependent on the laser power and the take-off angle (controlled by $\alpha$). Based on this equation, we obtained the energy densities at different laser power and irradiation positions, as shown in Supplementary Fig. 34a. Furthermore, according to the Newtonian equation:

$$Power\ density = Energy\ density/t_j = 1/2 \times mv_j^2/(m_{expanding} \times t_j) \qquad (9)$$

where $t_j$ is the actuation time. The power density of the hydrogel actuator can be estimated, as shown in Supplementary Fig. 34b. On the other hand, energy density and power density of the rolling behavior can be calculated based the following equations, respectively:

$$E_{rolling} = 1/2 \times mv_r^2 \qquad (10)$$

$$Energy\ density = E_{rolling}/m_{expanding} = 1/2 \times mv_r^2/m_{expanding} \qquad (11)$$

$$Power\ density = Energy\ density/t_r = 1/2 \times mv_r^2/(m_{expanding} \times t_r) \qquad (12)$$

where $t_r$ and $v_r$ are the actuation time and velocity of rolling, respectively. $m_{expanding}$ is the mass of the actuation part of the actuator (~0.21 mg, the same as that of jumping due to the similar bubble size). Note that $v_r$ is dependent on the laser power and irradiation position (controlled by $\alpha$). The estimated energy densities and power densities of rolling at different $\alpha$ and laser power are shown in Supplementary Fig. 35.

**Energy efficiency of the actuator**. The heat energy ($Q$) supplied by the light source can be estimated through the temperature change of the hydrogel actuator before and after light irradiation (Fig. 2h) based on the thermal energy equation[65]:

$$Q = mc\Delta T \qquad (13)$$

where $m$ is the mass of the irradiation area, which is ~0.21 mg; $\Delta T$ is the temperature difference of the hydrogel actuator before and after light irradiation, which is 126 °C; $c$ is the specific heat capacity of the hydrogel actuator.
For the hydrogel actuator, there are three main components, i.e., PAANa, IONP and water. Thus, the specific heat capacity ($c$) of the hydrogel actuator can be estimated to be ~2.5 kJ kg$^{-1}$ K$^{-1}$ by[66]:

$$c = \frac{mass_{PAANa} \times c_{PAANa} + mass_{IONP} \times c_{IONP} + mass_{water} \times c_{water}}{mass_{PAANa} + mass_{IONP} + mass_{water}} \qquad (14)$$

Therefore, Q can be obtained as 66.15 mJ.
On the other hand, the strain energy density caused by the bubble generation and hydrogel deformation is estimated by the numerical simulation (Supplementary Fig. 36). Since the model we simulated is symmetric, the strain energy can be estimated to be about 0.98 mJ by the integral of the energy over the whole volume. The energy efficiency can thus be calculated based on the following

equation to be 1.48%:

$$Energy\ efficiency = \frac{Strain\ energy}{Heat\ energy} = \frac{0.98\ mJ}{66.15\ mJ} = 1.48\% \qquad (15)$$

## Data availability

All data are available in the main text or the supplementary materials.

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

## Acknowledgements

This work was supported by the National Key Research and Development Program of China (Grant No. 2018YFE0306105), the National Natural Science Foundation of China (Grant No. 21574094), and the Collaborative Innovation Center of Suzhou Nano Science & Technology. It was also supported by the Priority Academic Program Development of Jiangsu Higher Education Institutions (PAPD), the 111 Project, Joint International Research Laboratory of Carbon-Based Functional Materials and Devices, the Fund for Excellent Creative Research Teams of Jiangsu Higher Education Institutions, the Post-graduate Research & Practice Innovation Program of Jiangsu Province (Grant No. KYCX19-1919), the China Scholarship Council (Grant No. 201906920034). M.S. was funded by the Max Planck Society and the European Research Council (ERC) Advanced Grant SoMMoR project with grant no: 834531. We also thank Prof. Y. Zhao and Prof. W. Chen for helpful discussions. Open access funding provided by Projekt DEAL.

## Author contributions

B.D., M.S., and M.L. conceived the presented idea, designed the research, and co-wrote the paper. M.L. and X.W. performed all experiments. All authors discussed the results and commented the manuscript.

## Competing interests

The authors declare no competing interests.
