## [Peer Review File · Nature Communications]

Reviewers' comments:

Reviewer #1 (Remarks to the Author):

The work presents the development and characterization of new light-driven hydrogel actuator, based on rapid phase transition (liquid to gas phase), able to jump and roll at a high speed and in air. The transition phase is generated by photothermal heating provided by a NIR laser mediated by iron oxide nanoparticles embedded in a hydrogel matrix. The authors show different motions and modality of control of the demonstrators. Respect to the state of the art of the field, the actuation here reported results impressively fast and reactive. The work has been clearly introduced and the level of innovation is clear and substantial. Finally the videos are very impressive and informative. The paper is well written, clear, and easy to follow. The language is appropriate and the paper is technologically sounding.

I have just a few comments that could improve the manuscript before the publication, as follows:

- Although in the methods it is explicitly reported that the "Unless otherwise mentioned, the light intensities to actuate the jumping and rolling motion of the hydrogel actuator were 1.2 and 0.3 W/cm², respectively", I think that this should be stated in the main text when the actuation is discussed, and in all the caption of the figures in which the motion is presented.

- The expression of laser light intensity in unit of [W/cm²] is not fully appropriate because, since the laser beam comes out from an optical fiber, the power strongly depends from the focus spot area. By the way this important parameter is not reported in the text. Probably it is not easy to define the area of IR spot, since the focalization seems to be made manually, but a range of spots areas should be provided. I suggest to use the power of the laser (expressed directly in W), which is the parameter that is controlled by the laser driver -and that can be more simply calibrated by standard procedures- and to calculate the typical power density (or light intensity) for selected applications, as reference for comparing the work with others.

- Other minor comment:

in Figure 1, panel "j" and "m" should report "jumping" and "rolling" directly in the graph, for a better readability.

"Among others, inspired by the motion of plants whose winding and curving are driven by the absorption or dehydration of water in the cells and tissues," insert reference(s) here.

Reviewer #2 (Remarks to the Author):

Li et al presented a new mechanism for hydrogel actuation and achieved ultra-high speed jumping and rolling by applying localized high-density NIR light. This is a very interesting work, overcoming the traditional drawbacks of hydrogel actuators.

I recommend its publication in NC. In addition, the following comments and concerns could be addressed to improve the paper.

1. Is this method limited to very small size beads (e.g., the bead with 1 mm in radius)? As shown in Equation 1, the driving force scales with the area, while the left part (gravity + mass times acceleration) scales with the volume. Therefore, it seems the driving force cannot necessarily be

high enough for a larger hydrogel bead. The largest size is 2.4 mm given by Fig. S12, which is still quite small in some real-world applications. The authors should point out the size and applicability of the technology.

2. Light controlled actuation of robots has the advantage of the remote control. But in the experiments showed by the authors, the light source needs to be very close to the beads to achieve efficient actuation. What is the difference between this light-assisted technology and manual operation of the beads?

3. Also, energy density and power density should be characterized.

4. As seen in the demonstrations of jumping and rolling, there is no way to control the trajectory or the specific destination of the actuator. Can the authors show the good controllability of the actuation trajectory or destination, without the help of a tunnel?

5. Besides, the authors only showed one-time actuation (or, "take-off"). How to achieve a sustainable, continuous motion? Since it is hard to predict the precise position of the destination, it is not very practical for the light to trace the beads after the first time actuation.

6. Hydrogel usually shows modulus of 1 kPa -1000 kPa. How can a hydrogel achieve a young's modulus of 100 MPa? (Supple Fig. 13)

7. In Figure 2m, error bars should be shown.

8. In the bubble formation, did the polymer network remain intact (reversible actuation), or fracture of hydrogel has happened (irreversible actuation)? Considering the brittleness of PAANA hydrogels, we would assume the gasification of water and bubble formation might cause the hydrogel fracture. More mechanical characterization and estimation of hydrogel fracture should be completed.

We would like to thank the Reviewers for their invaluable feedback and comments. We have addressed all comments outlined by the reviewers as detailed below point by point by significantly improving our paper. We have marked the reviewers' comments as *red italic* font and our responses as normal black font. The revised portion of the paper is highlighted in yellow in both the responses below and in the revised manuscript.

Reviewer 1:

The work presents the development and characterization of new light-driven hydrogel actuator, based on rapid phase transition (liquid to gas phase), able to jump and roll at a high speed and in air. The transition phase is generated by photothermal heating provided by a NIR laser mediated by iron oxide nanoparticles embedded in a hydrogel matrix. The authors show different motions and modality of control of the demonstrators. Respect to the state of the art of the field, the actuation here reported results impressively fast and reactive. The work has been clearly introduced and the level of innovation is clear and substantial. Finally, the videos are very impressive and informative.

The paper is well written, clear, and easy to follow. The language is appropriate and the paper is technologically sounding.

I have just a few comments that could improve the manuscript before the publication, as follows:

1. *'Although in the methods it is explicitly reported that the "Unless otherwise mentioned, the light intensities to actuate the jumping and rolling motion of the hydrogel actuator were 1.2 and 0.3 W/cm², respectively", I think that this should be stated in the main text when the actuation is discussed, and in all the caption of the figures in which the motion is presented.'*

Response: Thank you for your kind comment. We have revised the manuscript and provided the laser power in the main text and in all caption of the figures when the actuation is discussed according to your comment.

2. *'The expression of laser light intensity in unit of [W/cm²] is not fully appropriate because, since the laser beam comes out from an optical fiber, the power strongly depends from the focus spot area. By the way this important parameter is not reported in the text. Probably it is not ease to define the area of IR spot, since the focalization seems to be made manually, but a range of spots areas should be provided. I suggest to use the power of the laser (expressed directly in W), which is the parameter that is controlled by the lased driver -and that can be more simply calibrated by standard procedures- and to calculate the typical power density (or light intensity) for selected applications, as reference for comparing the work with others.'*

Response: Thank you for your comments.

i) The NIR laser beam is divergent and the spot area is dependent on the distance from the laser to the object. Because of the invisibility of the NIR light, we estimate the laser spot area by measuring the thermal image of the hydrogel actuator under NIR irradiation. As shown in new **Supplementary Figure 29(a)-(c)**, the NIR spot area increases with the increasing distance between the laser and the hydrogel actuator.

Furthermore, we have analyzed the above images utilizing the ImageJ software and obtained the NIR spot area under different circumstances, as shown in **Supplementary Figure 29(d)**. In the current study, the distance between the NIR laser and the hydrogel actuator is about 2 mm. The spot area is thus approximately 0.28 mm².

ii) In addition, we have revised the manuscript and used the power of the laser (W) instead of light density (W/cm²).

We have thus added the new **Supplementary Figure 29** and the corresponding descriptions are added on page 15, line 5 of the second paragraph: “The NIR laser beam without a collimator is divergent so that the spot area on the hydrogel actuator is dependent on the distance from the laser to the object and the temperature of the hydrogel actuator is only higher than 100 °C when the distance between the NIR light and the actuator is less than 4 mm (with a spot area of approximately 0.28 mm² at 2 mm separation) as shown in Supplementary Figure 29.”

Supplementary Figure 29. Infrared thermal images of the NIR (2.34 W) spot area on the surface of the hydrogel actuator. The distances between the laser and the hydrogel actuator are (a) 2 mm, (b) 7 mm and (c) 10 mm, respectively. (d) The NIR spot area as a function of the distance between the NIR laser and the hydrogel actuator.

3. *'In Figure 1, panel "j" and "m" should report "jumping" and "rolling" directly in the graph, for a better readability.'*

Response: Thank you for your comments. For clarity, we added “Jumping” and “Rolling” to Figures 1j and 1m, respectively in the revised Figure 1.

4. *“Among others, inspired by the motion of plants whose winding and curving are driven by the absorption or dehydration of water in the cells and tissues,” insert reference(s) here.’*

Response: Thanks for your comments. We have cited the corresponding reference as Reference 24 in the revised manuscript as:

(24) Ionov, L., Hydrogel-Based Actuators: Possibilities and Limitations. *Mater. Today* **2014**, *17*, 494.

Reviewer 2:

Li et al presented a new mechanism for hydrogel actuation and achieved ultra-high speed jumping and rolling by applying localized high-density NIR light. This is a very interesting work, overcoming the traditional drawbacks of hydrogel actuators.

I recommend its publication in NC. In addition, the following comments and concerns could be addressed to improve the paper.

1. *‘Is this method limited to very small size beads (e.g., the bead with 1 mm in radius)? As shown in Equation 1, the driving force scales with the area, while the left part (gravity + mass times acceleration) scales with the volume. Therefore, it seems the driving force cannot necessarily be high enough for a larger hydrogel bead. The largest size is 2.4 mm given by Supplementary Figure12, which is still quite small in some real-world applications. The authors should point out the size and applicability of the technology.’*

Response: Thank you for your kind comments.

- i) This method is not limited to the small-sized beads.

ii) The driving force originates from the shape deformation caused by the bubble formation due to the photothermal effect of Fe₃O₄ nanoparticles, which not only scales with the area but also scales with the contact pressure, as shown in the following equation (Equation 2 in the manuscript):

$$F_{jumping} = \sigma S$$

where S is the contact area between the hydrogel actuator and the substrate; σ is the contact pressure which can be obtained through the numerical simulation. σ is related to the pressure difference (ΔP) inside (P) and outside (P_0) the bubble, i.e. $\Delta P = P - P_0$, where P_0 is the atmospheric pressure and P is the pressure in the bubble cavity produced by the water vapor inside the hydrogel actuator. P can be expressed by the following equation (Equation 3 in the manuscript, J. Phys. Chem. Ref. Data 1993, 22, 783):

$$P = P_c \times e^{(T_c/T \times (a_1\tau + a_2\tau^{1.5} + a_3\tau^3 + a_4\tau^{3.5} + a_5\tau^4 + a_6\tau^{7.5}))}$$

where $P_c = 22.064$ MPa and $T_c = 647.096$ K are the critical pressure and temperature, respectively; T is the actual temperature of the hydrogel actuator; $\tau = 1 - T/T_c$; e and a_{1-6} are the constants. The simulated contact pressure (σ) depends on the size of the bubble cavity, as shown in new **Supplementary Figure 9**. The theoretical simulation indicates that the larger the bubble cavity leads to the higher contact pressure (σ) and thus the stronger driving force.

iii) In order to generate larger bubble, it is desirable to increase the illumination area while keeping the irradiation intensity. To this end, we have applied a collimator at the end of the NIR optical fiber, as illustrated in new **Supplementary Figure 28**. As a result, the jumping and rolling behaviors of big (8 mm sized) hydrogel actuator can be actuated by the NIR light, as shown in new **Supplementary Figure 32**.

iv) In recent years, small sized robotics (with sizes ranging from several hundreds of micrometers to several millimeters) have undergone tremendous progresses and hold great promises in the field of drug delivery, minimally invasive diagnosis and treatment inside the body, biotechnology, manufacturing or mobile sensor networks (Mobile Microrobotics, MIT

Press, Cambridge, MA, 2017). For the hydrogel actuator reported in the current study, its small size, fast response, rolling and jumping capability make it potentially attractive in the application field of cargo delivery, micromanufacturing and mobile sensor for environmental monitoring where the utilization of the large sized robots is inconvenient.

We have thus added these new supplementary figures as **Supplementary Figures 9, S28 and S32** and the corresponding descriptions are added on page 15, line 3 of the second paragraph: “To this end, it is desirable to keep the irradiation intensity by narrowing the divergent NIR beam through the application of a collimator at the end of the NIR optical fiber (Supplementary Figure 28).” And page 16, line 9 of the first paragraph: “In addition, the application of a collimator could help to generate bigger bubbles for bigger actuators. Since the driving force of the hydrogel actuator scales with the contact pressure (Equation 2) which depends on the size of the bubble cavity (Supplementary Figure 9), the bigger bubble formation provides the direct propulsion force for the jumping and rolling behaviors of big (8 mm sized) hydrogel actuator, as illustrated in Supplementary Figure S32, Supplementary Videos 23 and Video 24.” And on page 14, line 1 of the second paragraph: “In recent years, small sized robotics (with sizes ranging from several hundreds of micrometers to several millimeters) have undergone tremendous progresses and hold great promises in the field of drug delivery, minimally invasive diagnosis and treatment inside the body, biotechnology, manufacturing or mobile sensor networks.⁵⁹ The small size, fast response, rolling and jumping capability make the current actuator potentially attractive in different applications.”

Below are newly added supplementary figures for above changes:

Supplementary Figure 9. The contact pressure (σ) between the hydrogel actuator and the substrate versus the distance from the contact point between the hydrogel actuator and the substrate for different bubble cavity diameters (0.6 mm, 0.7 mm and 0.8 mm) obtained by simulation method. Note that the distance to the left is defined as negative.

Supplementary Figure 28. (a) Schematic showing the narrowing of the divergent NIR beam through the application of a collimator. (b) CCD image of the NIR optical fiber equipped with a collimator at the end.

Supplementary Figure 32. (a) Jumping (2.34 W irradiation) and (b) rolling (0.67 W irradiation) behavior of the big (8 mm sized) hydrogel actuator actuated by the NIR light (equipped with a collimator). (a) and (b) are overlaid images captured from Supplementary Video 23 and Video 24, respectively.

2. *'Light controlled actuation of robots has the advantage of the remote control. But in the experiments showed by the authors, the light source needs to be very close to the beads to achieve efficient actuation. What is the difference between this light-assisted technology and manual operation of the beads?'*

Response: Thank you for your comments.

i) The actuation of the hydrogel actuator relies on the bubble formation caused by the photothermal effect of Fe_3O_4 nanoparticle and the local temperature of the hydrogel actuator under NIR irradiation has to exceed $100\text{ }^\circ\text{C}$. However, the NIR light beam utilized in the current study is divergent. We have monitored the temperatures of the hydrogel actuator under NIR illumination when changing the distances between the NIR light and the actuator.

As can be seen from new **Supplementary Figure 29(a)-(c)**, the temperature of the hydrogel actuator is higher than 100 °C when the distance between the NIR light and the actuator is less than 4 mm. In order to increase the actuation distance, there are a few possible ways, which include the utilization of NIR laser with higher power, narrowing the divergent NIR light or their combination.

Among others, we have examined the possibility to increase the actuation distance by narrowing the divergent NIR beam utilized in the current study on the hydrogel actuator through the application of a collimator at the end of the NIR optical fiber, as shown in new **Supplementary Figure 28**.

We also monitor the temperature of the hydrogel actuator under the irradiation of the narrowed NIR light. As can be seen in **Supplementary Figure 30**, once the divergent NIR light is narrowed, the local temperature can still exceed 100 °C even if the distance between the light source and the actuator is increased to 35 mm.

As a result, the jumping and rolling behaviors of the hydrogel can be actuated by the NIR light when the distance between the light source (equipped with a collimator) and the actuator is 35 mm, as shown in new **Supplementary Figure 31**. Therefore, the actuation distance could be increased from < 4 mm before narrowing the divergent NIR light to ~35 mm after narrowing the NIR beam. It is anticipated that the actuation distance could be further increased when utilizing a NIR light source with higher power.

ii) There are some differences between light-assisted technology and manual operation, for example: a) the light-assisted technology is an untethered propelling method suitable for actuating the hydrogel actuator in a small and enclosed space (Mobile Microrobotics, MIT Press, Cambridge, MA, 2017) which cannot be achieved by the manual operation; b) the position of the laser light source could be precisely controlled by mechanical method through the utilization of a XYZ translation stage which is difficult to achieve for the manual operation method.

We have thus added these new supplementary figures as **Supplementary Figures 28, S30 and S31** and the corresponding descriptions are added on page 15, line 1 of the second paragraph: “In addition, we have studied the actuation of the current hydrogel actuator from a longer operation distance and the actuation of bigger sized ones, which may facilitate its further application in the practical field. To this end, it is desirable to keep the irradiation

intensity by narrowing the divergent NIR beam through the application of a collimator at the end of the NIR optical fiber (Supplementary Figure 28). The NIR laser beam without a collimator is divergent so that the spot area on the hydrogel actuator is dependent on the distance from the laser to the object and the temperature of the hydrogel actuator is only higher than 100 °C when the distance between the NIR light and the actuator is less than 4 mm (with a spot area of approximately 0.28 mm² at 2 mm separation) as shown in Supplementary Figure 29. Once the divergent NIR light is narrowed by the collimator, the local temperature can still exceed 100 °C even if the distance between the light source and the actuator is increased to 35 mm (Supplementary Figure 30). As a result, the jumping and rolling behaviors of the hydrogel can be actuated by the NIR light (2.34 W) (Supplementary Figure 31, Supplementary Video 21 and Video 22) at longer operation distance (35 mm), which is in contrast to < 4 mm without the application of a collimator. It is anticipated that the actuation distance could be further increased when utilizing a NIR light source with higher power.”

Newly added supplementary figures:

Supplementary Figure 30. Infrared thermal images of NIR light spot on the hydrogel actuator when the distance between the NIR optical fiber with a collimator end and the actuator is 35 mm under 2.34 W irradiation.

Supplementary Figure 31. (a) Jumping (2.34 W irradiation) and (b) rolling (0.67 W irradiation) behavior of the hydrogel actuator actuated by the NIR light (equipped with a collimator) which is placed 35 mm away from the actuator. (a) and (b) are overlaid images captured from Supplementary Video 21 and Video 22, respectively.

3. *‘Also, energy density and power density should be characterized.’*

Response: The light actuated shape deformation caused by the bubble formation due to the photothermal effect of IONPs produces energy ($E_{produced}$) that generates jumping ($E_{jumping}$) or rolling ($E_{rolling}$) motion by overcoming the stiction force to the ground ($E_{stiction}$), the air drag ($E_{air\ drag}$), the internal energy dissipation ($E_{dissipation}$), or the friction between the actuator and the ground ($E_{friction}$), as shown in the following (Soft Matter 2010, 6, 4342):

In case of jumping:

$$E_{produced} = E_{jumping} + E_{stiction} + E_{air\ drag} + E_{dissipation}$$

In case of rolling:

$$E_{produced} = E_{rolling} + E_{stiction} + E_{air\ drag} + E_{friction} + E_{dissipation}$$

The minimum produced energy could be estimated by $E_{jumping}$ (in case of jumping) or $E_{rolling}$ (in the case of rolling) because the produced energy ($E_{produced}$) is higher than that of jumping ($E_{jumping}$) or rolling ($E_{rolling}$).

Among others, $E_{jumping}$ can be analyzed based on the Newtonian equation as follows:

$$E_{jumping} = 1/2 \times mv_j^2$$

where m is the mass of the hydrogel actuator which is 7.95 mg; v_j is the take-off velocity. The energy density of the hydrogel actuator can thus be estimated by the following equation:

$$\text{Energy density} = E_{jumping} / m_{expanding}$$

where $m_{expanding}$ is the mass of the actuation part of the actuator (~0.21 mg), which is calculated from the size of the local expanding part ($m_{expanding} = \rho \times V_{expanding}$, where ρ is the density of the hydrogel actuator; $V_{expanding}$ is the volume of the local expanding part).

Therefore:

$$\text{Energy density} = 1/2 \times mv_j^2 / m_{expanding}$$

v_j is dependent on the laser power and the take-off angle (controlled by α which is defined as the angle between d (the virtual line between the irradiation position and the bottom of the actuator) and Z-axis (Supplementary Figure 16)). Based on this equation, we obtained the energy densities at different laser power and irradiation positions, as shown in new **Supplementary Figure 33(a)**.

Furthermore, according to the Newtonian equation:

$$\text{Power density} = \text{Energy density} / t_j = 1/2 \times mv_j^2 / (m_{expanding} \times t_j)$$

where t_j is the actuation time. The power density of the hydrogel actuator can be estimated, as shown in new **Supplementary Figure 33(b)**. The estimated power densities of jumping under light irradiation with different laser power and α .

On the other hand, energy density and power density of the rolling behavior can be calculated based the following equations, respectively:

$$E_{rolling} = 1/2 \times mv_r^2$$

$$\text{Energy density} = E_{rolling} / m_{expanding} = 1/2 \times mv_r^2 / m_{expanding}$$

$$\text{Power density} = \text{Energy density} / t_r = 1/2 \times mv_r^2 / (m_{expanding} \times t_r)$$

where t_r and v_r are the actuation time and velocity, respectively. $m_{expanding}$ is the mass of the actuation part of the actuator (~ 0.21 mg), which is calculated from the size of the local expanding part ($m_{expanding} = \rho \times V_{expanding}$, where ρ is the density of the hydrogel actuator; $V_{expanding}$ is the volume of the local expanding part). Note that v_r is dependent on the laser power and irradiation position (controlled by α). The estimated energy densities and power densities of rolling at different α and laser power are shown in new **Supplementary Figure 34**.

We have thus added the new **Supplementary Figures 33** and **S34** and the corresponding descriptions are added on page 18, line 1 of the second paragraph: “The light actuated shape deformation caused by the bubble formation due to the photothermal effect of IONPs produces energy ($E_{produced}$) that generates jumping ($E_{jumping}$) or rolling ($E_{rolling}$) motion by overcoming the stiction force to the ground ($E_{stiction}$), the air drag ($E_{air\ drag}$), the internal energy dissipation ($E_{dissipation}$), or the friction between the actuator and the ground ($E_{friction}$), as shown in the following:⁴⁹

In case of jumping:

$$E_{produced} = E_{jumping} + E_{stiction} + E_{air\ drag} + E_{dissipation}$$

In case of rolling:

$$E_{produced} = E_{rolling} + E_{stiction} + E_{air\ drag} + E_{friction} + E_{dissipation}$$

The minimum produced energy could be estimated by $E_{jumping}$ (in case of jumping) or $E_{rolling}$ (in the case of rolling) because the produced energy ($E_{produced}$) is higher than that of jumping

($E_{jumping}$) or rolling ($E_{rolling}$). Among others, $E_{jumping}$ can be analyzed based on the Newtonian equation as follows:

$$E_{jumping} = 1/2 \times mv_j^2$$

where m is the mass of the hydrogel actuator which is 7.95 mg; v_j is the take-off velocity. The energy density of the hydrogel actuator can thus be estimated by the following equation:

$$\text{Energy density} = E_{jumping} / m_{expanding}$$

where $m_{expanding}$ is the mass of the actuation part of the actuator (~0.21 mg), which is calculated from the size of the local expanding part ($m_{expanding} = \rho \times V_{expanding}$, where ρ is the density of the hydrogel actuator; $V_{expanding}$ is the volume of the local expanding part).

Therefore:

$$\text{Energy density} = 1/2 \times mv_j^2 / m_{expanding}$$

where v_j is dependent on the laser power and the take-off angle (controlled by α). Based on this equation, we obtained the energy densities at different laser power and irradiation positions, as shown in Supplementary Figure 33a. Furthermore, according to the Newtonian equation:

$$\text{Power density} = \text{Energy density} / t_j = 1/2 \times mv_j^2 / (m_{expanding} \times t_j)$$

where t_j is the actuation time. The power density of the hydrogel actuator can be estimated, as shown in Supplementary Figure 33b. On the other hand, energy density and power density of the rolling behavior can be calculated based the following equations, respectively:

$$E_{rolling} = 1/2 \times mv_r^2$$

$$\text{Energy density} = E_{rolling} / m_{expanding} = 1/2 \times mv_r^2 / m_{expanding}$$

$$\text{Power density} = \text{Energy density} / t_r = 1/2 \times mv_r^2 / (m_{expanding} \times t_r)$$

where t_r and v_r are the actuation time and velocity of rolling, respectively. $m_{expanding}$ is the mass of the actuation part of the actuator (~ 0.21 mg, the same as that of jumping due to the similar bubble size). Note that v_r is dependent on the laser power and irradiation position (controlled by α). The estimated energy densities and power densities of rolling at different α and laser power are shown in Supplementary Figure 34.”

Supplementary Figure 33. The estimated (a) energy densities and (b) power density of jumping at different laser power and α .

Supplementary Figure 34. The estimated (a) energy densities and (b) power densities of rolling at different laser power and α .

4. *'As seen in the demonstrations of jumping and rolling, there is no way to control the trajectory or the specific destination of the actuator. Can the authors show the good controllability of the actuation trajectory or destination, without the help of a tunnel?'*

Response: Thank you for your comments. In order to accurately control the trajectory of the actuator, we have fixed the NIR laser on an XYZ translation stage, as shown in new **Supplementary Figure 20**, and utilized a glass as the flat substrate and made it level with the assistance of a bubble level. We have studied the moving (jumping and rolling) trajectories and destinations of the hydrogel actuator when irradiating different locations on its surface using different laser power (the two factors that control the motion behavior of the hydrogel actuator).

For jumping, the laser is controlled to irradiate the bottom part of the hydrogel actuator with 0.39 W light irradiation and $\alpha = 70^\circ$ (α is defined as the angle between d (the virtual line between the irradiation position and the bottom of the actuator) and Z -axis), as shown in the following figure:

Schematic showing the irradiation position (red area) on the hydrogel actuator under 0.39 W light irradiation and $\alpha = 70^\circ$.

As can be seen from new **Supplementary Figure 21(a)**, which is obtained from **Supplementary Video 12** and the moving trajectory analysis from a number of actuators ($n = 30$) (b-c), the jumping trajectory and destination could be controlled at fixed laser power and α .

In addition, we have studied the jumping trajectory and destination by changing α . As shown in new **Supplementary Figure 23(a, b)**, the jumping trajectories and destinations are tunable by adjusting α through varying the irradiation position. Note that the irradiation is fixed at 0.39 W.

Furthermore, the jumping trajectory and destination of the actuator could also be adjusted by changing the laser power (α is fixed at 70°), as can be seen from new **Supplementary Figure 23(c, d)**.

These experiments indicate that the jumping trajectories and destinations of the hydrogel actuator are controllable by adjusting the laser power and the light irradiation position. Among others, the irradiation position and the laser power determine the take-off angle and the jumping height, respectively.

For rolling, the NIR light is controlled to irradiate the side part of the actuator with 0.14 W light irradiation and $\alpha = 60^\circ$, as illustrated in the following figure:

Schematic showing the irradiation position (red area) of the hydrogel actuator under 0.14 W light irradiation with $\alpha = 60^\circ$.

As can be seen from the overlaid image (a) shown in new **Supplementary Figure 22**, which is obtained from **Supplementary Video 13** and the rolling trajectory analysis from a number of actuators ($n = 30$) (b), the rolling trajectory and destination can be controlled when the laser power and irradiation position (α) are fixed.

Furthermore, we have studied the rolling destination by changing α and laser power. As shown in the following figure (a-b), the rolling destinations are controllable by adjusting α through varying the irradiation position (at fixed laser power, 0.14 W) or the laser power (at fixed $\alpha = 60^\circ$):

Rolling destination of the hydrogel actuator when changing (a) α (laser power is fixed at 0.14 W) or (b) the laser power (α is fixed at 60°).

The above experiments indicate that the actuator's trajectories and destinations can be controlled for both jumping and rolling behaviors when the irradiation position and laser power are fixed. And different trajectories and destination of the actuator could be realized by tuning the irradiation position or the laser power.

We have thus added new **Supplementary Figures 20-24** and the corresponding descriptions are added on page 12, line 1 of the third paragraph: “Furthermore, the actuator's trajectories and destinations can be controlled (Supplementary Figure 20) for both jumping and rolling behaviors when laser power and α are fixed. For example, the hydrogel actuator exhibits controllable jumping (Supplementary Figure 21 and Video 12) and rolling (Supplementary Figure 22 and Video 13) trajectory and destination under 0.39 W light irradiation/ $\alpha = 70^\circ$ and 0.14 W light irradiation/ $\alpha = 60^\circ$, respectively. In addition, different trajectories and destination of the actuator could be realized by tuning the irradiation position or the laser power. For instance, the actuator shows controllable jumping trajectories (take-off angle and the jumping height (Supplementary Figure 23 and Figure 19a)) and destinations and controllable rolling destination (Supplementary Figure 24 and Figure 19b) by adjusting α and laser power, respectively.”

Supplementary Figure 20. CCD camera image showing the experiment setup for the accurate control of the actuator motion by fixing the NIR laser on a XYZ translation stage.

Supplementary Figure 21. (a) Overlaid CCD image of the jumping behavior obtained from Supplementary Video 12 and (b) the trajectory analysis from a number of actuators ($n = 30$) under 0.39 W light irradiation and $\alpha = 70^\circ$. (c) The enlarged curve of (b) showing the initial trajectory of the jumping motion.

Supplementary Figure 22. (a) Overlaid CCD image obtained from Supplementary Video 13 and (b) the trajectory analysis from a number of actuators ($n = 30$) indicating the rolling behavior of the hydrogel actuator under 0.14 W light irradiation with $\alpha = 60^\circ$.

Supplementary Figure 23. (a) Jumping trajectories and destination of the hydrogel actuator under 0.39 W light irradiation when changing α . (b) The enlarged curve of (a) showing the initial trajectory of the jumping motion. (c) Jumping trajectories and destinations of the hydrogel actuator under different laser power (W) when α is fixed at 70° . (d) The enlarged curve of (c) showing the initial trajectory of the jumping motion.

Supplementary Figure 24. Rolling destination of the hydrogel actuator when changing α (laser power is fixed at 0.14 W).

5. *‘Besides, the authors only showed one-time actuation (or, “take-off”). How to achieve a sustainable, continuous motion? Since it is hard to predict the precise position of the destination, it is not very practical for the light to trace the beads after the first time actuation.’*

Response: Our response to question 4 indicates that the motion (jumping and rolling) trajectories and destinations of the hydrogel actuator can be predicted by adjusting the laser power and irradiation position. Therefore, by moving the laser source to the destination of the actuator after the first-time actuation, the second-time actuation could be realized. As shown in new **Supplementary Figure 25(a)** captured from Supplementary Video 14, we first actuate the hydrogel actuator to make it roll from position A to the designated position B. Then the laser is moved to position B to achieve the second-time actuation (jumping followed by rolling from position B to position C) in **Supplementary Figure 25(b)**.

Despite this controllability, we agree with the reviewer’s comments that it’s challenging for this hydrogel actuator to achieve the continuous actuation, which may be solved by the following: Directly mount a laser source on the hydrogel actuator through proper design so that the laser could move with the hydrogel actuator together, which may lead to the continuous actuation of the hydrogel actuator. Note that the capability to mount a battery

containing detectaphone on the hydrogel actuator has been demonstrated in **Supplementary Figure 26(b)**, implying the possibility to directly mount a NIR laser fiber (if not a laser source) on the actuator. Work along this direction is currently under progress. And in this paper, we mainly report an in-air hydrogel actuator with fast response and superior moving performance, which steps away from the water environment required for the previously reported hydrogel actuators.

We have thus added this figure as **Supplementary Figure 25** and the corresponding descriptions are added on page 13, line 5 of the first paragraph: “Based on this accurate control, the motion (jumping and rolling) destinations of the hydrogel actuator can be predicted. Therefore, by moving the laser source to the destination of the actuator after the first-time actuation, the second-time actuation could be achieved. As indicated in Supplementary Figure 25 captured from Supplementary Video 14, we first actuate the hydrogel actuator to make it roll from position A to the designated position B (under 0.14 W irradiation). Then the laser is moved to position B to achieve the continuous actuation (jumping followed by rolling from position B to position C, under 0.39 W irradiation).” Also, on page 15, line 5 of the first paragraph: “This also paves the way toward directly mounting a NIR laser fiber (even a laser source) on the actuator so that the laser could move with the hydrogel actuator together, which may lead to the continuous actuation of the hydrogel actuator.”

Supplementary Figure 25. Overlaid CCD image indicating (a) the rolling of the hydrogel actuator from position A to position B under 0.14 W irradiation and (b) jumping followed by

rolling from position B to position C under 0.39 W laser power. These images are captured from Supplementary Video 14.

6. 'Hydrogel usually shows modulus of 1 kPa -1000 kPa. How can a hydrogel achieve a young's modulus of 100 MPa? (Supple Figure 13)'

Response: In the current study, the Young's modulus of the hydrogel is measured by atomic force microscopy (AFM), as illustrated in the following figure:

Typical force curve of the hydrogel actuator containing IONPs obtained by AFM.

The Young's modulus in this case is 108 MPa. However, we also believe that the reviewer's opinion is correct. We have synthesized the stripe-shaped hydrogel and measured its stress-strain curve by utilizing the Instron tensile tester, as shown in the following figure:

Tensile stress-strain curve of the hydrogel with 2 wt% Fe_3O_4 content.

The Young's modulus in this case is calculated to be about 1.34 MPa, which is different from that obtained based on AFM measurement. Comparing the above two results, we think the Young's modulus measured by the tensile tester is more accurate due to the following reason:

The tensile tester measures the Young's modulus of the whole material, while AFM only measures the modulus of a small and localized surface. However, for the AFM measurement in the current case, the modulus of the hydrogel surface may be significantly higher than that of the bulk material because: During the AFM measurement, a NIR laser beam is applied which illuminates both the AFM tip and the IONP containing hydrogel. The NIR irradiation may cause the photothermal effect of IONP, leading to the dehydration of the pristine IONP containing hydrogel, the hardening of its surface and the increase of the Young's modulus.

Therefore, for accuracy, we adopt the tensile tester result instead of the AFM result. And the Young's modulus of the hydrogel with different IONP content measured by the tensile tester is shown in the figure below:

Young's modulus of the hydrogel with different IONP content.

We have thus added this figure as **Supplementary Figures 11c and 15** and the corresponding descriptions are added on page 17, line 6 of the fourth paragraph: “The Young’s modulus of the hydrogel comprising the actuator was obtained by utilizing a tensile tester (INSTRON-5942).” and page 10, line 7 of the first paragraph: “However, when the IONP content is too high, the Young's modulus of the resulting hydrogel increases dramatically (as shown in Supplementary Figure 15), reducing its flexibility and causing smaller bubble-induced deformation of the hydrogel actuator.”

7. *‘In Figure 2m, error bars should be shown.’*

Response: We have re-measured the weight of the hydrogel actuator before and after the actuation repeatedly and obtained the error bars in Figure 2m as shown in the following figure:

8. *'In the bubble formation, did the polymer network remain intact (reversible actuation), or fracture of hydrogel has happened (irreversible actuation)? Considering the brittleness of PAANA hydrogels, we would assume the gasification of water and bubble formation might cause the hydrogel fracture. More mechanical characterization and estimation of hydrogel fracture should be completed.'*

Response: We have performed more measurements to study whether the hydrogel fractures.

i) We have studied the surface morphology of the hydrogel containing 2 wt% IONP before and after NIR actuation (jumping and rolling) by utilizing the optical microscope. As

can be seen from new **Supplementary Figure 12**, negligible changes in the surface morphology are observed.

ii) Furthermore, we have synthesized the hydrogel comprising the actuator (containing 2 wt% IONP) in the shape of a stripe and studied its mechanical property by utilizing the Instron tensile tester. As can be seen from **Supplementary Figure 11**, the fracture of the hydrogel containing 2 wt% IONP occurs when the strain is higher than 42 %. According to the numerical simulation (the same simulation shown in Figure 2o in the manuscript), the maximum strain of the hydrogel actuator during bubble formation is estimated to be around 20%, which is much lower than the fracture strain of the hydrogel. Therefore, the hydrogel does not fracture during the bubble formation process, which also accounts for the re-usability of the hydrogel actuator (more than 100 times repetitive actuation, Supplementary Figure 11).

We have thus added this figure as **Supplementary Figures 11 and 12** and the corresponding descriptions are added on page 9, line 6 of the second paragraph: “According to the numerical simulation (Figure 2o), the maximum strain of the hydrogel actuator during bubble formation is estimated to be around 20%. This strain is lower than the fracture strain of the hydrogel comprising the actuator (i.e., > 42%, Supplementary Figure 11). This, together with the negligible changes in the surface morphology of the hydrogel comprising the actuator before and after NIR actuation (Supplementary Figure 12) and negligible water loss after the motion process (Figure 2m), indicates the hydrogel does not fracture during the bubble formation process, which also accounts for the re-usability of the hydrogel actuator (more than 100 times repetitive actuation, Supplementary Figure 13).”

Supplementary Figure 11. A stripe-shaped hydrogel containing 2 wt% IONP (a) before and (b) after stretching (42 % elongation). (c) Tensile stress-strain curve of the hydrogel containing 2 wt% IONP.

Supplementary Figure 12. Optical microscopic images showing the surface morphology of the hydrogel inside the hydrogel actuator after peeling of the surface PBR coating: (a) before and (b) after NIR actuation (2.34 W).

We hope that our responses and the revised manuscript are satisfactory to you and the reviewers. We will be happy to provide any further information if needed.

Sincerely,

Prof. Metin Sitti

Director, Max Planck Institute for Intelligent Systems

REVIEWERS' COMMENTS:

Reviewer #1 (Remarks to the Author):

Authors deeply revised the work following the reviewers' comments and remarks. They added interesting new explanations and technical details in the text, thus improving the quality of the manuscript.

Reviewer #2 (Remarks to the Author):

The authors have addressed my major concerns. Publication in NC of the paper is recommended. The authors may consider the following possible minor improvements or clarifications, which are not mandatory:

For the first question that I raised.

From Equation 2-3, I understand the relation between contact area and temperature. But what is the relation between contact pressure and bubble size? Could you directly give the analytic solution or simulated results of scaling relation between F-jumping and actuator size?

Why in Figure S9, the distance from the contact point between the hydrogel actuator and substrate was set as the x-axis for contact pressure, as we only care the initial time point when distance between hydrogel and substrate = 0?

I think the distance between the light source and the actuator as X-axis would make better sense, isn't it?

For the second question that I raised.

"in a small and enclosed space (Mobile Microrobotics, MIT Press, Cambridge, MA, 2017)" Could you give more practical & specific examples, that can manifest the advantages of your light assisted technology (distance ~35mm)?

For the third question that I raised.

I appreciate the authors analyzed the energy density from the "effect" side, that is, to estimate how much kinetic energy has been generated. It would be more informative if the authors can provide some energy analysis in terms of efficiency. For example, how much heat energy supplied by the light source, and the strain energy due to the bubble generation and hydrogel deformation. This would be helpful for the future optimization and provide design guidance for readers.

Response Letter

We would like to thank the reviewers for their invaluable feedback and comments. We have addressed all comments outlined by the reviewers as detailed below point by point which have significantly improved our paper. We have marked the reviewers' comments as *red italic* font and our responses as normal black font. The revised portion of the paper is highlighted with yellow color in both the responses below and in the revised manuscript.

Reviewer 1:

Authors deeply revised the work following the reviewers' comments and remarks. They added interesting new explanations and technical details in the text, thus improving the quality of the manuscript.

Response: Thank you for your kind comments.

Reviewer 2:

The authors have addressed my major concerns. Publication in NC of the paper is recommended. The authors may consider the following possible minor improvements or clarifications, which are not mandatory:

1. For the first question that I raised.

1) From Equation 2-3, I understand the relation between contact area and temperature. But what is the relation between contact pressure and bubble size? Could you directly give the analytic solution or simulated results of scaling relation between F-jumping and actuator size?

Response: The relation between contact pressure and bubble size is obtained through simulations. In order to successfully perform such simulations, different parameters, such as the bubble size, the actuator size, etc., need to be inputted ahead of time. In addition, as shown in the Supplementary Figure 9 and detailed response to the next comment, the contact pressure is different inside the contact area. Thus, we have studied the contact pressure at the center of the contact area (σ_{center}) as a function of the bubble size at the fixed actuator size (2 mm). As can be seen from the following figure, σ_{center} increases with the increasing bubble size.

σ_{center} as a function of the bubble size when the actuator size is fixed at 2 mm.

Similarly, the relation between σ_{center} and the actuator size could be obtained through the simulation method by changing the actuator size when the bubble size is fixed (0.6 mm), as shown in the following figure:

σ_{center} as a function of the actuator size when the bubble size is fixed at 0.6 mm.

In addition, the relation between $F_{jumping}$ and the actuator size could be obtained based on equation 2, i.e. $F_{jumping} = \sigma S$, as shown in the following:

$F_{jumping}$ as a function of the actuator size when the bubble size is fixed at 0.6 mm.

We thus added the above figures to the supplementary information as Supplementary Fig. 32 and the corresponding descriptions are added on page 36 of the Supplementary information:

“The relation between contact pressure and bubble size is obtained through the simulation method. In order to successfully perform the simulation, different parameters, such as the bubble size, the actuator size, etc., need to be inputted ahead of time. In addition, as shown in the Supplementary Figure 9, the contact pressure is different inside the contact area. Thus, we have studied the contact pressure at the center of the contact area (σ_{center}) as a function of the bubble size (at the fixed actuator size, 2 mm) and the actuator size (when the bubble size is fixed at 0.6 mm) through the simulation method. As can be seen from (a-b), σ_{center} increases with the increasing bubble size and actuator size. The relation between $F_{jumping}$ and the actuator size could be obtained based on equation 2, as shown in (c).” And on page 16, line 11 of the first paragraph in the manuscript: “Since the driving force of the hydrogel actuator scales with the contact pressure (Equation 2), which depends on the size of the bubble cavity and the hydrogel actuator (Supplementary Fig. 9 and Fig. 32), the larger bubble formation would provide a direct propulsion force for the jumping and rolling behaviors of large (e.g., 8 mm diameter) hydrogel actuators, as illustrated in Supplementary Fig. 33, Video 23 and Video 24.”

Supplementary Figure 32. (a) σ_{center} as a function of the bubble size when the actuator size is fixed at 2 mm. (b) σ_{center} and (c) $F_{jumping}$ as a function of the actuator size when the bubble size is fixed at 0.6 mm.

- 2) *Why in Figure S9, the distance from the contact point between the hydrogel actuator and substrate was set as the x-axis for contact pressure, as we only care the initial time point when distance between hydrogel and substrate = 0? I think the distance between the light source and the actuator as X-axis would make better sense, isn't it?*

Response: Thank you for your comments. Please see the following illustration for the force analysis:

(a) The illustration showing the numerical simulation and (b) side view and (c) top view of the contact area.

According to equation 2: $F_{jumping} = \sigma S$, the force depends on S , i.e. the contact area, which is illustrated in the above figure as the green colored area. d is the distance from the center of contact to a certain position inside the contact area ((b) in the above figure). And the contact pressure (σ) inside the contact area is different at different location, i.e. the contact pressure is the highest at the center (σ_{strong} , (b) in the above figure), medium in the middle (σ_{medium} , (b) in the above figure) and the smallest at the edge (σ_{weak} , (b) in the above figure). The following figure (Supplementary Figure 9) shows the typical contact pressure distribution inside the contact area as a function of d . And the total force inside the contact area is obtained by the integral of the contact pressure over the entire contact area ((c) in the above figure).

Supplementary Figure 9. The contact pressure (σ) versus the distance from the center of contact (d) between the hydrogel actuator and the substrate for different bubble cavity diameters (0.6 mm, 0.7 mm and 0.8 mm) obtained by the simulation method. Note that the distance to the left is defined as negative.

We have thus added these figures to Supplementary Fig. 9 and the corresponding descriptions are added on page 13 of the Supplementary information:

“According to equation 2: $F_{jumping} = \sigma S$, the force depends on S , i.e. the contact area, which is illustrated in (I -III) as the green colored area. d is the distance from the center of contact to a certain position inside the contact area (II -III). And the contact pressure (σ) inside the contact area is different at different location, i.e. the contact pressure is the highest at the center (σ_{strong}), medium in the middle (σ_{medium}) and the smallest at the edge (σ_{weak}). (IV) shows the typical contact pressure distribution inside the contact area as a function of d . Therefore, the total force can thus be obtained by the integral of the contact pressure over the entire contact area (III).”

Supplementary Figure 9. (I) The illustration showing the numerical simulation and (II) side view and (III) top view of the contact area. (IV) The contact pressure (σ) versus the distance from the center of contact (d) between the hydrogel actuator and the substrate for different bubble cavity diameters (0.6 mm, 0.7 mm and 0.8 mm) obtained by the simulation method. Note that the distance to the left is defined as negative.

2. *For the second question that I raised.*

*“in a small and enclosed space (Mobile Microrobotics, MIT Press, Cambridge, MA, 2017)”
Could you give more practical & specific examples, that can manifest the advantages of your light assisted technology (distance ~35mm)?”*

Response: The hydrogel actuator reported in this study may have many potential applications in robotics, biomedical and miniature device application. We exemplify two possible practical/specific application scenarios. First, the current actuator may be utilized for

unclogging or repairing the jammed or damaged pipes and tubings made of NIR-transparent materials, such as plastics and glasses, by releasing cargoes inside. Such repair/maintenance industrial application is possible for space shuttles, nuclear plants, airplanes, and other vehicles or infrastructural buildings, where there is limited access to small pipes/tubes (see new refs. 61 and 62). Next, by utilizing biodegradable components, the resulting actuator with small sizes (e.g., 100 μm , Supplementary Figure 14f) may achieve the active and precise delivery of drugs (due to its controlled motion behavior driven by NIR light) for disease or blood clot treatment after injection to the blood vessels close to skin (within 10 mm depth) on legs, arms or neck (see new refs. 63 and 64). Our actuator's materials can be easily biodegradable (already iron oxide nanoparticles are biodegradable; we just need to make the hydrogel biodegradable, which is straightforward). An NIR light source from outside would control the motion of the hydrogel ball and release the drug at the target location.

We thus added the new text on page 17:

“Specifically, the current actuator may be utilized for unclogging or repairing jammed or damaged pipes or tubes made of NIR-transparent materials, such as plastics and glasses, by releasing specific cargoes inside. Such industrial repair/maintenance application is possible for nuclear plants, space shuttles, airplanes, etc., where there is limited access to small pipes/tubes.^{61,62} In addition, by utilizing biodegradable hydrogels, small-scale (e.g., 100 μm diameter, Supplementary Figure 14f) hydrogel balls may achieve the active and precise delivery of drugs (due to its controlled motion behavior driven by external NIR light) for disease or blood clot treatment after injection to the blood vessels close to skin (within \sim 10 mm depth) on legs, arms or neck.^{63,64}”

3. *For the third question that I raised.*

I appreciate the authors analyzed the energy density from the “effect” side, that is, to estimate how much kinetic energy has been generated. It would be more informative if the authors can provide some energy analysis in terms of efficiency. For example, how much heat energy supplied by the light source, and the strain energy due to the bubble generation and hydrogel deformation. This would be helpful for the future optimization and provide design guidance for readers.’

Response: On the one hand, we estimate the heat energy (Q) supplied by the light source through the temperature change of the hydrogel actuator before and after light irradiation as shown in the following figure (Figure 2h).

The temperature change during the motion process.

According to the following thermal energy equation (Renew. Sust. Energ. Rev. 2009, 13, 318):

$$Q = mc\Delta T$$

where m is the mass of the irradiation area, which is approximately 0.21 mg; ΔT is the temperature difference of the hydrogel actuator before and after light irradiation which is 126 °C; c is the specific heat capacity of the hydrogel actuator.

For the hydrogel actuator, there are three main components, i.e. poly(sodium acrylate) (PAANa), iron oxide nanoparticle ($\text{Fe}_3\text{O}_4\text{NP}$) and water. Thus, the specific heat capacity (c) of the hydrogel actuator can be estimated by the following equation (J. Mater. Sci. Mater. Med. 2004, 15, 1061) to be approximately $2.5 \text{ kJ kg}^{-1} \text{ K}^{-1}$. Therefore, Q can be obtained which is 66.15 mJ.

$$c = \frac{\text{mass}_{PAANa} \times c_{PAANa} + \text{mass}_{Fe_3O_4NP} \times c_{IONP} + \text{mass}_{water} \times c_{water}}{\text{mass}_{PAANa} + \text{mass}_{Fe_3O_4NP} + \text{mass}_{water}}$$

On the other hand, the strain energy density caused by the bubble generation and hydrogel deformation is estimated by the simulation process as shown in the following figure:

The strain energy density of the shape deformation of the hydrogel actuator obtained by using the numerical simulation.

Since the model we simulated is symmetric, the strain energy can be estimated to be about 0.98 mJ by the integral of the energy over the whole volume. The energy efficiency can thus be calculated based on the following equation to be 1.48%:

$$\text{Energy efficiency} = \frac{\text{Strain energy}}{\text{Heat energy}} = \frac{0.98 \text{ mJ}}{66.15 \text{ mJ}} = 1.48\%$$

We added the above figure to the supplementary information as Supplementary Fig. 36 and the corresponding descriptions are added on page 20:

“Energy efficiency of the actuator. The heat energy (Q) supplied by the light source can be estimated through the temperature change of the hydrogel actuator before and after light irradiation (Fig. 2h) based on the thermal energy equation.⁶⁵

$$Q = mc\Delta T \quad (13)$$

where m is the mass of the irradiation area, which is approximately 0.21 mg; ΔT is the temperature difference of the hydrogel actuator before and after light irradiation which is 126 C; c is the specific heat capacity of the hydrogel actuator.

For the hydrogel actuator, there are three main components, i.e. poly(sodium acrylate) (PAANa), iron oxide nanoparticle ($\text{Fe}_3\text{O}_4\text{NP}$) and water. Thus, the specific heat capacity (c) of the hydrogel actuator can be estimated to be approximately $2.5 \text{ kJ kg}^{-1} \text{ K}^{-1}$ by:⁶⁶

$$c = \frac{\text{mass}_{\text{PAANa}} \times c_{\text{PAANa}} + \text{mass}_{\text{Fe}_3\text{O}_4\text{NP}} \times c_{\text{IONP}} + \text{mass}_{\text{water}} \times c_{\text{water}}}{\text{mass}_{\text{PAANa}} + \text{mass}_{\text{Fe}_3\text{O}_4\text{NP}} + \text{mass}_{\text{water}}} \quad (14)$$

Therefore, Q can be obtained as 66.15 mJ.

On the other hand, the strain energy density caused by the bubble generation and hydrogel deformation is estimated by the numerical simulation (Supplementary Fig. 36). Since the model we simulated is symmetric, the strain energy can be estimated to be about 0.98 mJ by the

integral of the energy over the whole volume. The energy efficiency can thus be calculated based on the following equation to be 1.48%:

$$\text{Energy efficiency} = \frac{\text{Strain energy}}{\text{Heat energy}} = \frac{0.98 \text{ mJ}}{66.15 \text{ mJ}} = 1.48\% \quad (15)$$

Supplementary Figure 36. The strain energy density of the shape deformation of the hydrogel actuator obtained by using the numerical simulation.

We hope that our responses and the revised manuscript are satisfactory to you and the reviewers. We will be happy to provide any further information if needed.

Sincerely,

Prof. Metin Sitti
Director, Max Planck Institute for Intelligent Systems